# Food Safety and Waste Management in TV Cooking Shows: A Comparative Study of Turkey and the UK

**DOI:** 10.3390/foods14152591

**Published:** 2025-07-24

**Authors:** Kemal Enes, Gülbanu Kaptan, Edgar Meyer

**Affiliations:** 1Vocational School, Tarsus University, Mersin 33400, Türkiye; kemalenes@tarsus.edu.tr; 2Leeds Business School, University of Leeds, Leeds LS2 9JT, UK; e.meyer.1@bham.ac.uk; 3Birmingham Business School, University of Birmingham, Birmingham B15 2TT, UK

**Keywords:** food safety, food waste, behaviour, TV cooking shows, MasterChef, video content analysis

## Abstract

This study examines food safety and waste behaviours depicted in the televised cooking competition *MasterChef*, a globally franchised series that showcases diverse culinary traditions and influences viewers’ practices. The research focuses on the *MasterChef* editions aired in Turkey and the United Kingdom, two countries with distinctly different social and cultural contexts. Video content analysis, based on predefined criteria, was employed to assess observable behaviours related to food safety and waste. Additionally, content analysis of episode transcripts identified verbal references to these themes. Principal Component Analysis was employed to categorise patterns in the observed behaviours. The findings revealed frequent lapses in food safety, with personal hygiene breaches more commonly observed in *MasterChef UK*, while cross-contamination issues were more prevalent in *MasterChef Turkey*. In both versions, the use of disposable materials and the discarding of edible food parts emerged as the most common waste-related practices. These behaviours appeared to be shaped by the cultural and culinary norms specific to each country. The study highlights the importance of cooking shows in promoting improved food safety and waste management practices. It recommends involving relevant experts during production and clearly communicating food safety and sustainability messages to increase viewer awareness and encourage positive behaviour change.

## 1. Introduction

The issue of food waste is an important global concern. The United Nations 2024 Food Waste Index report reveals that 1052 billion tonnes of food waste were generated in 2022, with 60% occurring in homes. Previous research indicates that food waste patterns differ among nations with different income levels [1,2,3]. For instance, the annual per capita household food waste in Turkey was reported as 74 kg [4] in 2022 compared to 95 kg in the United Kingdom [5].

Ensuring food safety is essential worldwide to protect citizens from foodborne illnesses, promote good health and quality of life, and prevent the economic burden that these illnesses place on healthcare systems and businesses due to lost productivity. According to the World Health Organisation [6], an estimated 600 million people become ill from consuming contaminated food each year, and 420,000 die. The economic impact associated with productivity losses and medical expenses amounts to approximately USD 110 billion annually. In the UK alone, an estimated 2.4 million people suffer from foodborne illnesses each year, leading to around 180 deaths. Contrary to the common belief, most foodborne illnesses in the UK occur at home rather than outside of the home [7]. Despite ongoing efforts by the food safety and waste organisations, there is still a need to improve citizens’ behaviours regarding food safety [8,9,10] and waste [11,12]. Research has indicated that TV cooking shows can negatively influence viewers’ food safety and food waste behaviours by sometimes failing to provide accurate information [8,13]. Moreover, these shows often do not sufficiently emphasise the importance of proper food safety practices and waste prevention strategies [14]. Cooking shows have also been suggested as a means to increase consumer awareness and promote food safety practices [15]. Therefore, we argue that these shows can serve as effective interventions to positively impact viewers’ food safety and waste behaviours.

Sezerel & Filimonau highlight that cultural traits, social norms, and moral attitudes influence chefs’ behaviours regarding food waste reduction [16]. As the MasterChef cooking shows are broadcast in countries with different social and cultural backgrounds and culinary heritages, we argue that food safety and waste behaviours of the chefs in these shows may vary, affecting viewers differently. For example, it has been reported that British chefs in upscale restaurants possess a strong understanding of food waste and its negative socioeconomic and environmental impacts [17]. They implement strategies such as monitoring, composting, reusing ingredients, menu planning, and food donation to minimise waste [18]. On the other hand, Turkish people are particularly mindful of avoiding food and drink waste [19] due to their religious beliefs. For example, traditional Turkish cuisine encompasses a range of dishes and cooking techniques that aim to minimise food waste, including the strategic use of spices to enhance the shelf life of cooked foods [20,21].

Our study focuses on Turkey and the UK, each with its own unique social, cultural, and culinary practices, and both airing the *MasterChef* cooking show. We assessed chefs’ behaviours on *MasterChef* TV programmes to develop recommendations for TV producers and celebrity chefs, with the aim of encouraging them to positively influence viewers’ food safety and waste behaviours. To our knowledge, no other study has examined both food safety and waste behaviours of the chefs in cooking shows to date.

### 1.1. Food Safety and Waste Practices in Turkey and the UK

In a study conducted in Turkey, 16.5% of participants reported having had food poisoning due to improper food safety behaviours, such as using expired food, improper storage conditions, and contaminated food [10]. A recent study involving Turkish women aged 18 to 77 years revealed that participants in the 18–29 and 30–41 age groups exhibited inadequate safety practices, such as defrosting frozen foods at room temperature. Additionally, the study indicated that, across most educational levels, participants demonstrated similar poor food safety practices [22]. In the UK, it was found that not washing hands, especially after handling raw food, along with cross-contamination from inadequate surface cleaning, contributed to foodborne diseases. Additionally, many consumers in the UK mistakenly consider convenience meal products as “low-risk” for foodborne illnesses, while damaged packaged products are often a source of such diseases [23]. A cross-national study examining consumer awareness of food safety in Turkey, Germany, and Hungary showed a significantly higher level of awareness among German participants compared to those from Turkey and Hungary [24]. Additionally, it was found that limited food safety knowledge among Turkish participants was significantly correlated with educational and income levels [25]. Furthermore, Bolek suggested that foodborne illnesses in Turkey may be reduced if media communications, such as television, effectively promote food safety behaviours [26].

Research shows that households significantly contribute to food waste, and the behaviours are influenced by various cultural practices, which manifest in differing social norms, traditions, and expectations [27]. The relationship between culture and food waste is complex and multifaceted. In China, cultural values, such as maintaining dignity and conforming to group norms, have a significant impact on young consumers’ food waste behaviours, particularly when dining out [28]. Additionally, factors such as materialism and individualism influence consumers’ ethical standards regarding food waste [29].

Notably, individuals who hold strong ethical or religious objections to wasting food often face challenges in making shopping lists and meal planning but excel in managing excess food [30]. Cultural and religious beliefs can also promote the prevention of food waste. For instance, women in rural Lebanese households exhibit a strong aversion to food waste, rooted in cultural norms and religious beliefs, despite not being particularly motivated by environmental concerns in their food-related practices [31]. This highlights the intricate interplay between cultural values, religious beliefs, and food waste behaviour. Understanding the cultural context is essential for developing effective strategies to reduce food waste. Adapting sustainable practices to fit cultural traditions can help ensure food security, while embracing cultural diversity can foster effective and sustainable food consumption patterns globally. These positive changes can be implemented if people are willing to modify their behaviours [32,33]. Therefore, the aim of our study is to identify the differences in food waste and safety behaviours of chefs from Turkey and the UK.

### 1.2. Impact of TV Cooking Shows on Viewers

In recent years, celebrity chefs have become national icons and trendsetters. With their unique culinary talents and captivating personalities, they have the power to inspire, educate, and influence people worldwide [34]. Thus, they have started to branch out from the kitchen and engage in different pursuits [35]. Celebrity chefs have become writers, entrepreneurs, role models, and TV stars [36]. Woods indicates that celebrity chefs have become role models for culinary students; however, many of their shows often violate food safety rules, which can have a negative impact on students [37].

Television cooking programmes featuring celebrity chefs are on the rise. In a study conducted by [38], 50% of consumers in the USA reported watching cooking shows very often or occasionally. Additionally, 57% of these viewers reported shopping for food based on meal suggestions provided in these cooking programmes. Kızılırmak and Demir found that 52.1% of Turkish participants who watched cooking programmes gained more knowledge about recipes and cooking methods [39]. Furthermore, 22.7% of the participants applied recipes they learnt from these programmes to improve their cooking skills at home. It has been suggested that improving cooking skills helps reduce household food waste [40]. Moreover, incorporating food safety information into cooking recipes has shown promise in improving consumers’ food safety behaviours, such as hand washing and the use of thermometers [15]. These findings suggest that cooking programmes may positively influence audience behaviour [9,41], thereby supporting food waste reduction and encouraging proper food safety practices in households.

Borda et al. conducted a study in the UK to evaluate how effectively TV cooking shows promote safe food handling practices [42]. The authors analysed 19 popular cooking programmes across Europe using principal content analysis. They found that most of the programmes do not adequately address food safety. Geppert et al. showed that hygiene- or food safety-related mistakes were detected every 50 s in a German TV cooking show [43]. Güner and Şimşek demonstrated that several important food safety practices, such as not wearing excessive make-up or elaborate hairstyles, not licking spoons to taste food, and maintaining a clean workspace, are often ignored in these shows [44]. Additionally, celebrity chefs featured in various programmes were found to frequently fail to demonstrate proper food safety behaviours. Some of these behaviours had the potential to lead to cross-contamination [15]. These studies indicate that the behaviours of chefs on TV cooking shows may contribute to foodborne illnesses if replicated by viewers at home [9,38,45].

Mettke suggests that *MasterChef Germany* highlights the importance of waste separation and offers advice on reducing food waste [46]. However, *MasterChef U.S.* does not follow this approach, as the judges often throw away “inedible” and “bad” food on the floor, contributing to food waste. In another study, three types of cooking shows were analysed: a competition show, a morning live programme, and a famous programme by a Turkish chef. While these shows provided helpful information for the viewers, the study concluded that there is still room for improvement [47]. The communication and behaviours in the cooking shows can influence household food waste positively or negatively and thus may have a significant impact on food waste [8]. The research questions of our study are as follows:How frequently are food safety principles violated in television cooking programmes broadcast in Turkey and the UK?How often are food-waste behaviours observed in television cooking programmes broadcast in Turkey and the UK?What are the similarities and differences between cooking programmes in the UK and Turkey regarding food safety violations and behaviours for preventing food waste?What are the implications of our findings for promoting positive behaviour change through TV cooking shows?

## 2. Materials and Methods

This research first employs a qualitative approach to identify negative food safety and food and kitchen waste behaviours on TV cooking shows in the UK and Turkey. Video content analysis (VCA) and content analysis (CA) were used to assess behaviours. Content analysis is a technique that is usually employed to analyse textual data but can also be used to analyse other forms of data, such as videos or pictures. The application of content analysis to videos and pictures is referred to as multimedia content analysis [48]. or video content analysis [49].

Content analysis involves categorising and condensing data [50]. Our study examines food safety and food waste behaviours in the TV cooking show *MasterChef*. This show was chosen for our analysis due to its broad broadcast range, popularity, and high viewer ratings, as well as the fact that its episodes are publicly available and easily accessible for viewing. Ethical approval for this study was obtained from Tarsus University (Turkey) and the University of Leeds (UK). As the videos were accessed through publicly available platforms, no additional consent was required. *MasterChef* has been aired in nearly 100 countries worldwide, including the United Kingdom and Turkey. *MasterChef UK* is regarded as “the third most popular all-time food and drinks TV Programme” [51] and has been on air since 2005 [52]. Meanwhile, *MasterChef Turkey* holds the top spot among Turkish TV shows [53]. The competition features contestants competing to demonstrate their culinary skills, and at the end of each season, the strongest performer is crowned the *MasterChef* winner.

We assessed the food safety-related content of the cooking shows based on the criteria developed by Borde et al. [42]. Table 1 presents these items, representing the categories of personal hygiene (A), cross-contamination (B), proper cooking (C), storage conditions, and risk management (D).

The first and second authors of this study, who have expertise in culinary sciences and food waste-related behavioural decision research, developed a second evaluation scale to determine improper food waste practices in cooking shows. This scale was based on the criteria established by the Waste and Resources Action Programme (WRAP) [54]. To validate the scale, 19 culinary experts from the UK and 20 experts from Turkey reviewed and evaluated it. The experts included celebrity chefs hosting TV cooking shows, executive chefs from 5-star chain hotels and fine dining establishments and Michelin-starred restaurants, as well as chefs teaching gastronomy and culinary arts programmes in academic institutions. Strict ethical standards were maintained to protect participant confidentiality and data integrity. The data were anonymised and securely stored in compliance with relevant data protection regulations. The ethical protocols were reviewed and approved by the institutional review boards of the institutions of the first author.

For the evaluation process, the culinary experts were tasked with identifying appropriate criteria for assessing TV cooking shows. They were asked to indicate their level of agreement (i.e., agree or disagree) for each criterion. To determine the final list of items, we employed Lawshe’s Content Validity Ratio (CVR) [55]. The CVR is calculated using the formula CVR $=\frac{n_{e}-(N/2)}{N/2}$, where ‘ne’ represents the number of agreeing experts and ‘N’ denotes the total number of experts involved. In our study, which included 39 experts, the CVR value needed to exceed 0.29, with at least 25 experts required to express agreement [56]. The two items with CVR values below 0.29 were eliminated: “Paying attention to saving energy” and “Reusing stale food (e.g., bread crust, breadcrumbs) or dairy products.” However, two additional items, despite having CVR values lower than 0.29, were retained because more than 60% of experts agreed on them, and their CVR values were close to the threshold of 0.29. These items are “Wasting food because of lack of knowledge about processing techniques (e.g., losing meat juices during cooking, damaging vegetables because of improper cutting) and “Choosing packaged food that would lead to waste.” The complete list of items evaluated by the experts, along with the percentage of agreement for each item, the content validity ratio (CVR), and the final decision for each item, is presented in Table 2.

As a result of the evaluation process, nine out of the eleven items were retained on the list. However, the item “Using only the loin, sirloin and the other prime, choice or selective cut of the meat and using only some parts of the veggies or fruits “was split into two distinct items to separately reflect vegetables and meat, following recommendations from several experts. The final list of items used to assess the *MasterChef* episodes in relation to food waste criteria is presented in Table 3.

Initially, Principal Component Analysis (PCA) was employed to verify that the scales used were appropriately categorised before testing, ensuring that the outcomes aligned with these categories. PCA and factor analysis techniques help examine a single group of variables, identifying which ones form consistent subgroups that are relatively autonomous from one another. By extracting and analysing the underlying structure of the data, PCA facilitates the interpretation of complex datasets, allowing researchers to gain insights into the relationships between variables and to reduce dimensionality while retaining essential information. This process is crucial for ensuring the validity and reliability of the research findings.

### Statistical Analysis

All statistical analyses were conducted using IBM SPSS Statistics (Version 29.0, 2023). Lawshe’s Content Validity Ratio (CVR) was calculated manually to assess the relevance of each item in the food waste behaviour scale. Inter-rater reliability for the video content analysis was evaluated using Cohen’s Kappa, ensuring consistency between coders.

Principal Component Analysis (PCA) was applied to the coded data to explore the underlying structure of the behavioural categories. PCA was conducted using varimax rotation, and components were retained based on eigenvalues greater than 1 and a visual inspection of the scree plot. No inferential statistical tests were performed; however, descriptive information such as the number of episodes analysed, total cooking time, and the number of transcribed dialogues and sentences was reported to contextualise the dataset.

## 3. Results and Discussion

### 3.1. Video Content and Descriptive Content Analysis

An inter-rater reliability analysis was conducted with NVivo 14 to evaluate the consistency of observations within the video content analysis. Two independent coders assessed the same video segments, and their observations were compared. For food safety behaviours, the analysis yielded an average Cohen’s kappa value of 0.94 and an agreement rate of 99.95%. For waste behaviours, the average kappa value was 0.91, with an agreement rate of 99.91%. A kappa value of 0.70 or higher is generally considered acceptable as an indicator of inter-rater reliability [56].

To conduct the descriptive content analysis, we first transcribed all episodes of *MasterChef UK* and *Turkey*. This process resulted in 29,138 dialogues, 31,288 sentences, and 152,105 words for *MasterChef UK* and 27,953 dialogues, 77,312 sentences, and 358,054 words for *MasterChef Turkey*. For the analysis of the transcripts, we employed an interpretive content analysis approach [57] using MAXQDA 2022 PRO.

All 21 episodes of *MasterChef UK* and 21 episodes (out of 177) of *MasterChef Turkey* that were broadcast in 2022 were visually analysed. *MasterChef UK’s* 21 episodes were 1217 min and 30 s in duration. Each of these six episodes (i.e., numbers 3, 6, 9, 12, 15, and 18) lasted approximately 28 min. The remaining episodes had an average duration of 58 min each. In these longer episodes, a total of 547 min and 10 s was dedicated to presentation and evaluation, while 423 min and 57 s were spent on cooking. Additionally, 246 min and 23 s were allocated for introductions and the evaluation of the dishes. The total duration of the 21 episodes of *MasterChef Turkey* lasted 3423 min and 47 s, with each episode averaging 163 min. The final episode had the longest runtime, at 240 min. Of the total time, 2177 min and 29 s were spent on foretalk, presentation, and evaluation, while 1198 min and 29 s were dedicated to actual cooking. Moreover, 47 min and 49 s were used for introductions.

Regarding food safety, we identified 267 additional unsafe behaviours that were not part of the evaluation scale we used [42]. These behaviours ranged from disorganised workspaces and damaging Teflon cookware with metallic utensils to handling ready-to-serve dishes with unclean hands. Examples of risky behaviours included using cleaning cloths meant for worktops to wipe competitors’ sweat, consuming cooked meat without utensils, and neglecting food safety guidelines. These findings are consistent with previous research [38,42,46], particularly regarding violations of personal hygiene and cross-contamination. Personal hygiene violations were observed more frequently on *MasterChef UK*, occurring every 57 s, compared to every 2 min and 32 s on *MasterChef Turkey*. A comprehensive list of these observations is provided in Appendix A. We note that these behaviours were not included in our analysis, as they were not included in Borde et al.’s evaluation scale [42].

#### 3.1.1. Video Content Analysis Results on Food and Kitchen Waste

We recorded 821 behaviours in *MasterChef UK* and 1073 in *MasterChef Turkey* across 42 episodes. In *MasterChef UK*, there were 365 instances of wasting behaviours, occurring on average every 66 s, while *MasterChef Turkey* had 562 instances, with occurrences every 28 s. The number of waste observations and corresponding time intervals are shown in Appendix A.

*Edible food waste.* In *MasterChef UK*, we recorded 101 observations every 24 min and 58 s, compared to 48 observations in *MasterChef Turkey*. Of the observations made in *MasterChef UK*, 45 were related to the question, “Did they use only the loin, sirloin, and other prime, choice, or selective cuts of meat?” This was specifically within the “Seafood or poultry” category. Research indicates that the percentage of an animal’s weight that is suitable for human consumption varies by species [58]. For example, the yields are 55% for cattle and 66% for poultry, indicating that a 300 kg calf yields approximately 165 kg of edible meat, including only 3 kg from the loin and 18 kg from the sirloin [59,60]. This focus on premium cuts often results in the underutilisation of other parts of the animal.

In contrast, *MasterChef Turkey* used premium cuts sparingly, featuring them only four times, as Turkish cuisine typically utilises all parts of the animal. This approach reflects the emphasis on minimising waste, rooted in nomadic traditions [61] and religious principles. Additionally, the versatility of certain dishes, such as kebabs and meatballs, supports the use of various animal parts [62]. Food waste, resulting from limited knowledge of processing techniques and the use of only certain parts of fruits and vegetables, was evident in nearly every episode in both countries.

*Inedible kitchen waste.* This category focuses on inedible kitchen items such as aluminium foil, stretch film, and baking paper. In *MasterChef UK*, there were 173 observations every 2 min and 27 s, compared to 356 observations every 3 min and 22 s in *MasterChef Turkey*. Although *MasterChef Turkey* had more observations, the frequency of these observations was lower than that of *MasterChef UK*.

The most frequently observed behaviour involves the use of aluminium foil, baking paper, or stretch film, with 141 observations every 3 min in *MasterChef UK* and 265 observations every 4 min and 31 s in *MasterChef Turkey*. The remaining observations pertain to packaged food. One study found that only 45% of household waste is biowaste; the rest consists of combustible materials, chemical paper waste, glass, aluminium, steel, and other similar items, including kitchen waste [63].

Additionally, the selection of packaged food that contributes to waste was noted twice in each episode in both countries. The only exception was the behaviour of “avoiding the use of excessive amounts of water or liquid,” which was not observed in *MasterChef UK*. This absence may be attributed to the shorter episode duration of *MasterChef UK*, which averages 58 min, compared to the typical 2 h and 43 min for *MasterChef Turkey*.

*Avoidable food waste*. This category includes innovative strategies for repurposing food by-products, such as the use of eggshell powder as a calcium supplement in chocolate cake preparation [64]. In this context, *MasterChef UK* recorded 91 relevant observations every 4 min and 40 s, while *MasterChef Turkey* recorded 158 observations every 7 min and 35 s in response to the question, “Did they waste unused parts of the food?” The majority of these observations involved the disposal of edible components, such as outer layers of onions and skins of tomatoes and root vegetables.

#### 3.1.2. Video Content Analysis Results on Food Safety

In 21 episodes of *MasterChef UK*, there were 446 instances of food safety violations and 10 positive behaviours, with unsafe ones occurring every 57 s during cooking. In *MasterChef Turkey*, 474 food safety violations and 37 positive ones were noted, with unsafe behaviours observed every 2 min and 32 s. The number of all food safety violations and corresponding time intervals are presented in Appendix A.

A frequent observation in *MasterChef Turkey* was the use of separate chopping boards for raw meat and seafood. This practice reflects distinct characteristics shaped by the food cultures and culinary traditions of each country. For instance, Turkish cuisine includes specific preparations for kebabs that involve minced meat, requiring the use of a specialised cleaver and a dedicated cutting board.

Direct contact with food, particularly raw meat, can contaminate hands and surfaces with pathogens such as *E. coli*, which can lead to severe foodborne illnesses [65,66]. An analysis of *MasterChef UK* revealed that food safety and personal hygiene violations occurred every 2 min and 26 s, resulting in 174 incidents. In contrast, *MasterChef Turkey* had 91 such incidents every 13 min and 10 s. Interestingly, handwashing was noted only twice in the UK version and once in the Turkish version. Most risky behaviours involved wearing jewellery or watches, with 129 instances in *MasterChef UK* (every 3 min and 17 s) and 90 in *MasterChef Turkey* (every 13 min and 19 s). In a recent study conducted across ten European countries, the UK sample was ranked as seventh (preceding Portugal, France, and Spain) with a lower likelihood of washing hands after touching raw chicken and percentage scores for proper hand cleaning methods and key moments for hand washing [67]. Another study conducted with UK food handlers following the COVID-19 pandemic suggested that frequent hand hygiene failures among food handlers cannot be attributed to knowledge gaps, as the main safety message communicated by the UK government during the pandemic was *Hands. Face. Space*, along with their well-publicised official guidance of *Wash hands for 20 s* [68]. Therefore, we argue that the less rigorous handwashing behaviours of the *MasterChef UK* contestants compared to their Turkish counterparts may result from differing social norms rather than a knowledge gap among the UK participants.

In *MasterChef UK*, contestants with long nails and nail polish were observed 42 times every 10 min and 6 s. In contrast, such observations were not recorded in *MasterChef Turkey*, likely due to the jury’s clear stance against wearing jewellery and nail polish, as noted in episode 156. Research indicates that long nails, if not properly disinfected, can harbour microorganisms [69]. Additionally, jewellery carrying pathogens such as Staphylococcus and Streptococcus may lead to cross-contamination [70]. Therefore, nails and jewellery that are not properly cleaned can pose significant health risks.

Our research identified 272 behaviours that compromise food safety in *MasterChef UK*, occurring every minute and 34 s. In *MasterChef Turkey*, we found 383 behaviours contributing to cross-contamination every three minutes and 8 s. In *MasterChef UK*, the most common problem was not washing hands after handling raw meat and seafood, with 109 instances noted every 3 min and 53 s. In contrast, *MasterChef Turkey* had only 25 instances of this behaviour every 47 min and 56 s. The main concern in *MasterChef Turkey* was not using separate chopping boards for different foods, occurring every 3 min and 533 s, totalling 337 instances. This is concerning, as chopping boards are a major source of cross-contamination [57].

A significant concern regarding cooking shows is the frequency of food safety recommendations. In *MasterChef UK*, there were 10 instances of food safety recommendations, whereas *MasterChef Turkey* featured 37 such recommendations. However, these positive examples were greatly outnumbered by instances of improper behaviour. Research suggests that viewers of cooking programmes are likely to adopt the food safety practices they observe [37]. Consequently, there is a need for *MasterChef* to have more food safety behaviours and provide recommendations during the shows.

In addition to the limited number of food safety recommendations, *MasterChef UK* episodes revealed several instances of unsafe food handling practices. Contestants were observed using the same knives after handling raw meat without washing them and frequently licking the tasting spoons. These observations, combined with the broader lack of emphasis on safe cooking practices and storage conditions, prompted us to investigate further. To address this, we conducted a descriptive content analysis, employing an interpretive approach as recommended in the literature [71].

#### 3.1.3. Descriptive Content Analysis Results on Food Safety and Waste

The content analysis identified 99 instances of food safety and waste behaviours in *MasterChef UK* and 52 in *MasterChef Turkey*. Most of the safety behaviours were associated with the questions “Did they receive any instruction about cooking time”? and “Did they assess whether the meat or fish was cooked properly”? To analyse these instances, we adopted an interpretive approach, focusing on evaluative phrases such as “It appears to be undercooked,” “It seems that it has not cooked enough”, “I think that it should be cooked a little bit more”, and “It needs to cook 15 more minutes”. These expressions were recorded 61 times in *MasterChef UK* and 17 times in *MasterChef Turkey*. The lower frequency of such comments in the Turkish episodes may be attributed to the contestants’ tendency to prepare traditional Turkish dishes, which are often served slightly overcooked [61,72,73,74]. In contrast, *MasterChef UK* featured a noticeable trend toward serving rare liver, a practice associated with an increased risk of *Campylobacter* infections [75]. These findings suggest that cooking-related safety issues are more prevalent in *MasterChef UK*, highlighting the need for television cooking programmes to prioritise the communication of safe cooking principles.

Waste-related behaviours also emerged as a significant theme in the content analysis, particularly in relation to the use of leftovers. In *MasterChef Turkey*, food waste was actively discouraged in 14 instances, with an additional 7 moments promoting the reuse of leftover ingredients. Examples of such messaging include statements like, “This is a zero-waste plate”, “Do not waste anything from fish”, and “We won’t throw anything away; we’re using every part of the food”. One judge even remarked, “Our only request is that you store the unused parts in the refrigerator so that they don’t go to waste, because wasting them would be a sin”. Notably, a guest chef, recipient of the 2023 Basque Culinary World Prize, was invited in one episode specifically to advocate for waste reduction. In contrast, *MasterChef UK* featured less communication around food waste. Only one episode addressed the issue, focusing on the use of shrimp heads and other typically discarded parts. While the official *MasterChef UK* website outlines a strong commitment to sustainability and food waste prevention [52], no explicit messaging on this topic was observed in any episode of the 2022 UK season. In comparison, *MasterChef Turkey* highlighted food waste prevention 13 times during the same period.

This disparity may be partly explained by cultural and religious influences. Due to Islamic teachings that emphasise the importance of avoiding waste [20,21], Turkish audiences tend to be particularly mindful of food and drink conservation [19]. Traditional Turkish cuisine also reflects this ethos, incorporating a variety of techniques and dishes designed to minimise waste. For example, spices are often used to extend the shelf life of cooked foods [20,21]. Leftover ingredients, such as stale flatbreads, day-old bread, and excess cooked meat or vegetables, are commonly repurposed into dishes like *börek*, *papara*, and *bread mantı* [19]. These culinary practices not only reflect cultural values but also offer practical models for sustainable cooking that could be more widely promoted in televised cooking programmes.

### 3.2. Principal Component Analysis

To investigate the underlying factors associated with food waste and unsafe food handling behaviours, Principal Component Analysis (PCA) was employed [49] with IBM SPSS. This method, in conjunction with factor analysis techniques, was used to examine a unified set of variables with the objective of identifying coherent groupings. These groupings, or components, represent clusters of related behaviours that are internally consistent yet relatively independent from one another.

#### 3.2.1. Food and Kitchen Waste Behaviours

The principal component analyses revealed three factors for both countries. A cut-off value of “0.45” was applied during the factor analysis to ensure significant correlations [48]. The results are summarised in Table 4. The PCA scree plots illustrating waste behaviours in *MasterChef Turkey* and *MasterChef UK* are shown in Figure 1. Following the Kaiser criterion, we focused on the first three components that had eigenvalues greater than one, as these components captured the most significant variation in waste behaviours and were retained for further analysis.

Table 4 shows the PCA results, highlighting significant findings in bold. These results reveal the underlying dimensions of waste behaviours. Three distinct factors were extracted, each representing a coherent grouping of related behaviours. The first factor, labelled “Knowledge”, reflects behaviours associated with a lack of awareness regarding food handling and waste. In *MasterChef Turkey*, strong factor loadings were observed for the items “wasting food due to lack of knowledge about processing techniques” (0.779) and “choosing packaged food that would lead to waste” (0.847 Similarly, in *MasterChef UK*, the items “wasting food due to lack of knowledge about processing techniques” (0.861) and “discarding prime cuts of meat” (0.892) exhibited strong loadings on this factor, indicating a shared underlying dimension. These behaviours often occur during the early stages of food preparation and reflect individual decision-making rather than technical constraints.

The second factor, labelled “Inedible Kitchen Waste”, captures behaviours involving the disposal of materials generally considered non-edible. In the Turkish dataset, “using wooden skewers, aluminium foil, baking paper, stretch film, etc.” (0.784) and “wasting the unused parts of the food” (0.818) showed strong factor loadings. In the UK dataset, “wasting the unused parts of the food” also showed a strong correlation (0.738), suggesting a consistent pattern across contexts. These behaviours are typically observed during cooking or preparation and reflect choices that increase the amount of non-edible kitchen waste. The third factor, identified as “Avoidable Food Waste,” highlights behaviours that result in the unnecessary disposal of edible components. In *MasterChef Turkey*, “using only some parts of vegetables or fruits” (0.803) and “avoiding the use of excessive amounts of water or liquid” (0.714) were the primary contributors. In *MasterChef UK*, the items “produce-sorting techniques” (0.697) and “using wooden skewers, aluminium foil, baking paper, stretch film, etc.” (−0.855) loaded on this factor, the latter indicating a negative association with avoidable waste. These behaviours often occur during ingredient selection or final preparation and reflect personal preferences or habitual practices.

Overall, while the factor structures exhibit thematic similarities across both datasets, notable differences also emerge. These may reflect cultural or contextual variations in food handling and waste behaviours. The UK dataset demonstrated a clearer separation of factors and a higher total variance explained (81.77%) compared to the Turkish dataset (71.49%), suggesting more distinct behavioural clustering in the UK sample.

#### 3.2.2. Food Safety Behaviours

The PCA results and PCA scree plots are demonstrated in Table 5 and Figure 2, respectively. We identified two factors that had eigenvalues greater than one for each country. The first factor in the analysis of *MasterChef Turkey consisted* of two items, whereas the UK version included three. Notably, the cleaning process was not observed in *MasterChef UK*, and long or polished nails were not present in *MasterChef Turkey*. However, all three observations indicate risky food practices. The only difference is that wearing jewellery is associated with a different factor in Turkey compared to the UK. Jewellery can be a personal hygiene concern that may lead to contamination.

The second factor had three items in both countries. These items were primarily related to raw meat and cross-contamination, including hand washing after touching raw meat or seafood, using different chopping boards for raw meat or seafood, and changing or washing knives after contact with raw meat or seafood. Notably, changing or washing knives after contact with raw meat or seafood was not observed in *MasterChef Turkey*. We labelled the first factor as “hygiene and cleaning” and the second factor as “cross-contamination”.

The factor analysis of food safety paralleled the waste analysis, utilising similar categorisation and scales. These findings also reaffirmed the reliability of the scales.

## 4. Conclusions

This study examined the prevalence of improper food safety and waste management behaviours in a televised cooking show broadcast in Turkey and the United Kingdom, with the aim of informing strategies that can positively influence viewers’ behaviours at home.

Our findings highlight the important role of cooking shows in shaping public attitudes toward food safety and sustainability. While entertainment remains their primary function, these programmes also carry a responsibility to model safe and environmentally responsible food practices.

As the first cross-national and mixed-methods analysis of food safety and waste behaviours in a prime-time cooking show, our research makes a novel contribution to both media studies and sustainability education. It offers a new perspective for understanding the cultural and social dynamics of televised food media.

We recommend that media producers collaborate with food safety and sustainability experts to ensure accurate, culturally sensitive, and responsible messaging. By embedding these values into both production processes and on-screen content, shows like *MasterChef* can positively contribute to public health, environmental stewardship, and cultural awareness. Importantly, our analysis highlights how cultural and social differences influence culinary practices and, by extension, food safety and waste behaviours.

Rather than assigning blame, this research highlights the opportunity for cooking shows to serve as platforms for positive behavioural change. By aligning entertainment with education and cultural sensitivity, producers can help foster healthier, more sustainable food practices among a diverse audience.

## Figures and Tables

**Figure 1 foods-14-02591-f001:**
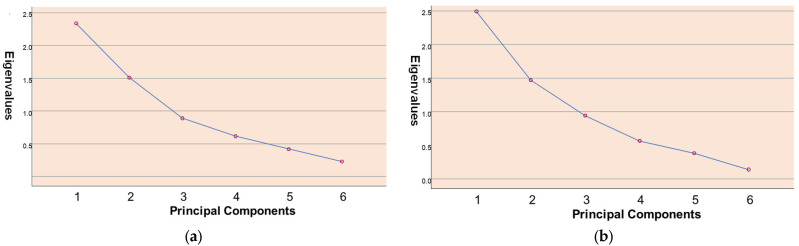
(**a**) Scree plot of principal component analysis on waste behaviours in *MasterChef Turkey*. (**b**) Scree plot of principal component analysis on waste behaviours in *MasterChef UK*.

**Figure 2 foods-14-02591-f002:**
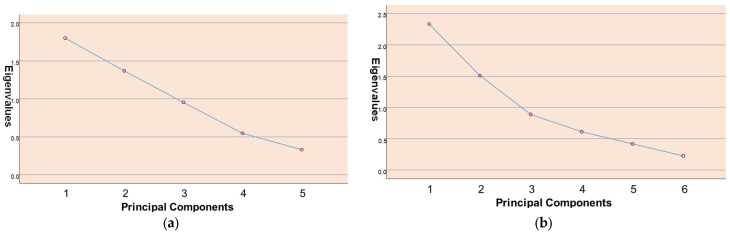
(**a**) Scree plot of principal component analysis on food safety behaviours in *MasterChef Turkey.* (**b**) Scree plot of principal component analysis in *MasterChef UK*.

**Table 1 foods-14-02591-t001:** Items in the food safety evaluation scale. Source: Borda et al. [43].

Category	Items
A	Washing hands before starting to cook.
	Wearing a uniform or any similar protection equipment.
	Wearing jewellery and/or a watch.
	Having long or/and polished nails.
B	Washing their hands after touching raw meat or fish.
	Using different chopping boards for raw meat or fish.
	Changing knives or washing them after contact with raw meat/fish.
	Changing spoons or washing properly after tasting the cold meal.
C	Assessing whether meat or fish was cooked properly (by using a thermometer).
	Giving instructions about cooking time.
	Giving instructions about cooking temperature.
	Giving instructions about cooling temperature.
	Giving instructions about cooling time.
D	Providing food safety information.
	Providing contradictory food safety recommendations.
	Mentioning cleaning procedures.

**Table 2 foods-14-02591-t002:** The list of items evaluated by culinary experts (N = 39), the % of agreement, CVR values, and final decision for each item.

	Percent Agreement (%)	Content Validity Ratio (CVR)	Decision
Wasting unused parts of the food (e.g., Potato skin, onion skin)	89.7	0.79	Included
Wasting food because of lack of knowledge about processing techniques (e.g., losing meat juices during cooking, damaging vegetables because of improper cutting)	61.5	0.23	Included
Using only the loin, sirloin and the other prime, choice or selective cut of the meat and using only some parts of the veggies or fruits (e.g., using the green parts of the onions)	66.7	0.33	Included but split into two
Reusing stale food (e.g., bread crust, breadcrumbs) or dairy products (e.g., milk, cheese)	59.0	0.18	Excluded
Choosing packaged food that would lead to waste	61.5	0.23	Included
Using wooden skewers, aluminium foil, baking paper, stretch film, etc.	79.5	0.59	Included
Wasting any tea bags or coffee grounds	71.8	0.44	Included
Wasting single-use kitchen materials (e.g., single-use cups, plastic dishes)	71.8	0.44	Included
Paying attention to saving energy	56.4	0.13	Excluded
Avoiding unnecessary use of detergents (e.g., using the dishwasher)	66.7	0.33	Included
Avoid using excessive amounts of water or liquid	74.4	0.49	Included

**Table 3 foods-14-02591-t003:** The list of items used to assess food waste behaviours in TV shows.

Category	Item
Edible Waste	Wasting food because of lack of knowledge about processing techniques
	Using only the loin, sirloin and the other prime, choice or selective cut of the meat
	Using only some parts of the veggies or fruits
Kitchen Waste	Choosing packaged food that would lead to waste
	Using wooden skewers, aluminium foil, baking paper, stretch film, etc.
	Wasting single-use kitchen materials
	Avoiding unnecessary use of detergents
	Avoiding using excessive amounts of water or liquid
Avoidable Waste	Wasting unused parts of the food (e.g., potato skin, onion skin, eggshell)
	Wasting tea bags or coffee grounds

**Table 4 foods-14-02591-t004:** Results of the Principal Component Analysis of food waste behaviours.

	Turkey			UK		
	F1: Knowledge	F2: Inedible Kitchen Waste	F3: Avoidable Food Waste	F1: Knowledge	F:2 Inedible Kitchen Waste	F:3 Avoidable Food Waste
Did they waste food because of their lack of knowledge about processing techniques?	**0.779**	−0.088	0.361	**0.861**	0.154	0.215
Did they choose packaged food that would lead to waste?	**0.847**	0.074	−0.139	−0.086	**0.950**	−0.038
Did they use wooden skewers, aluminium foil, baking paper, stretch film, etc.?	0.285	**0.784**	0.121	−0.005	0.261	**−0.855**
Did they waste the unused parts of the food?	−0.252	**0.818**	−0.133	**0.581**	**0.738**	−0.020
Did they use only some parts of the veggies or fruits?	0.016	0.261	**0.803**	0.311	0.344	**0.697**
Did they avoid using excessive amounts of water or liquid?	0.089	−0.328	**0.714**	-----	-----	-----
Did they use only the loin, sirloin and the other prime, choice, or selective cuts of the meat? (Seafood or poultry)	-----	-----	-----	**0.892**	−0.050	0.056
% Variance explained	%28.97	%24.45	%18.07	%41.53	%24.52	%15.72
Cumulative % variance	%28.97	%53.42	%71.49	%41.53	%66.05	%81.77

Extraction Method: Principal Component Analysis. Rotation Method: Varimax with Kaiser Normalisation. Total Variance Explained: 71.49% (Turkey), 81.772% (UK).

**Table 5 foods-14-02591-t005:** Results of the principal component analysis of food safety behaviours.

	Turkey		UK	
	F1: Hygiene and Cleaning	F2: Cross-Contamination	F1: Hygiene and Cleaning	F2: Cross-Contamination
Did they wear jewellery and/or a watch?	0.119	**−0.631**	**0.728**	0.388
Did they wash their hands after touching raw meat or seafood?	0.261	**0.796**	0.088	**0.829**
Did they use a different chopping board for raw meat or seafood?	−0.415	**0.571**	0.229	**0.734**
Were the knives changed or washed after contact with raw meat/seafood?	-------	-------	−0.054	**0.553**
Were the spoons changed or washed properly after tasting the cold or hot meal?	**0.882**	0.160	**0.785**	−0.348
Were any cleaning procedures mentioned?	**0.860**	−0.231	**-------**	**------**
Have they long or polished nails	-------	-------	**0.889**	0.207
% Variance explained	%35.99	%27.35	%38.92	%25.169
Cumulative % variance	%35.99	%63.34	%38.92	%64.09

Extraction Method: Principal Component Analysis. Rotation Method: Varimax with Kaiser Normalisation.

## Data Availability

The original data presented in the study are openly available in the Open Science Framework at https://osf.io/sftxz/?view_only=545a4e91b11f4dbe89504d214e83784f (accessed on 23 April 2025).

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
