# Peer review of "Food Safety and Waste Management in TV Cooking Shows: A Comparative Study of Turkey and the UK"

_foods, 2025, doi:10.3390/foods14152591_

Round 1
Reviewer 1 Report
Comments and Suggestions for Authors
Introduction: The flow of the introduction needs to be improved and shortened. Avoid repetition. Include lines 412-419 in the introduction to motivate the choice of including Turkey in the study. It is necessary to motivate the use of the UK and Turkey in the study when MasterChef TV programs are also produced in other countries.
Language: Employ an English Language editor to assist in the language, eliminating repetition and improving the flow of the paper.
Word use: Could "wooden screws" be replaced with "wooden skewer"?
In line 480: The researcher used different data analysis methods, but could it technically be described as a "mixed-method approach"?
More explanation is necessary for some of the "behaviors". For instance "excessive use of water" - what is meant by "excessive"? Also, "waste of inedible materials" - was that only counted when such materials were wasted or when such materials were used. These are only two examples, however overall more explanation is needed for the exact parameters of what was regarded as waste and food security misbehaviors.
The results section and discussion sections seems to contain too much repetition of results. Perhaps the two sections could be incorporated into each other to form a more concise presentation of the results.
The conclusion should be more concise, to summarize and bring into context with the impact on society. The recommendations and other discussions should become part of the main body of text.
Comments on the Quality of English LanguageIt is suggested that the authors be given the opportunity to improve the paper by employing an English language editor to correct the use of words, flow and avoid repetition.
Author Response
Comments 1 Introduction: The flow of the introduction needs to be improved and shortened. Avoid repetition. Include lines 412-419 in the introduction to motivate the choice of including Turkey in the study. It is necessary to motivate the use of the UK and Turkey in the study when MasterChef TV programs are also produced in other countries.
Response 1: Thanks for pointing this out. We structured the introduction of our original manuscript in the following order: The food waste problem at global, UK, and Turkey levels; the food safety problem at global, UK, and Turkey levels; the importance of cooking shows in influencing viewer behaviour; aim of our study and the research gap it addresses; and the rationale for focusing on the UK and Turkey in our study. While we believe the flow of our original introduction is sound, we have still improved it to address the reviewer’s comments. Additionally, we have shortened the introduction (please see the deleted lines, which are crossed out and marked in red in p.2 of the revised manuscript) and added a similar version of the text on lines 412-419 into the introduction of the revised manuscript as below (also see page 2, lines 71-75).
|
On the other hand, the Turkish people are particularly mindful of avoiding food and drink waste [19] due to their religious beliefs. For example, traditional Turkish cuisine encompasses a range of dishes and cooking techniques that aim to minimise food waste, including the strategic use of spices to enhance the shelf life of cooked foods [20,21]. |
Comments 2 Language: Employ an English Language editor to assist in the language, eliminating repetition and improving the flow of the paper.
Response 2: In response to your comment, we have carefully revised the paper to improve clarity, eliminate repetition, and enhance overall readability. We have used Grammarly as a language support tool during this process to help identify and correct issues related to grammar, style, and coherence.
Comments 3 Word use: Could "wooden screws" be replaced with "wooden skewer"?
Response 3: Thank you for pointing this out. We have replaced all “screws” with “skewers” in the revised manuscript (see p. 6, Table 2; p. 7, Table 3; p.12, Table 4, lines 467 and 547). Additionally, we have replaced all “aluminium folios” with “aluminium foil” in the revised manuscript (marked in red, e.g., p.6, Table 2)
Comment 4: In line 480: The researcher used different data analysis methods, but could it technically be described as a "mixed-method approach"?
Response 4: Because our content and video content analysis involved qualitative interpretation and our PCA was used to analyse patterns quantitively, we believe we can refer to our approach as a mixed-methods approach (i.e., the integration of both qualitative and quantitative research methods in a single study to provide a more comprehensive understanding of a research problem).
However, to address the Reviewer’s comment and as we have revised the results and discussion largely, we have not mentioned about mixed-method approach in the revised results and discussion section of the manuscript.
Comment 5: More explanation is necessary for some of the "behaviors". For instance "excessive use of water" - what is meant by "excessive"? Also, "waste of inedible materials" - was that only counted when such materials were wasted or when such materials were used. These are only two examples, however overall more explanation is needed for the exact parameters of what was regarded as waste and food security misbehaviors.
Response 5: Regarding the term “excessive use of water,” we intended it to refer to situations where more water was used than necessary. For example, in Episode 116 of MasterChef Turkey, some contestants left the tap running even after completing the rinsing of raw fish. To improve clarity, we have revised the term to “excessive amounts of water” throughout the manuscript (see p.6 Table 2; p.7 Table 3; p.12 Table 4; and p.12, line 474 in the revised manuscript).
As for “waste of inedible materials,” we could not locate this exact phrase in the manuscript. However, we assume the reviewer is referring to “inedible waste,” which in our study denotes inedible kitchen items such as aluminium foil, stretch film, and baking paper. This was defined in the original submission (p.10, lines 318–319). To enhance clarity, we have revised the term to “inedible kitchen waste” in the updated manuscript (see p.8, line 314; p.11 Table 4; and p.12, line 465).
We believe the remaining behavioural categories are sufficiently explained in the manuscript. Additionally, the language editing applied during revision should have improved overall clarity. However, if there are specific behaviours that still require further elaboration, we would be happy to provide additional clarification if requested, ideally with page and line references
Comment 6: The results section and discussion sections seems to contain too much repetition of results. Perhaps the two sections could be incorporated into each other to form a more concise presentation of the results.
Response 6: Thank you for this suggestion. To address the suggestions of Reviewers 1 and 3, we have incorporated the results and discussion sections and removed the repetitions. We have also made the PCA Tables more informative and clearer, and relocated Tables 4 and 5 (in the original manuscript) to the Supplementary Information to address the comments from Reviewers 2 and 4. We believe that revising the PCA tables and relocating two tables to the supplementary information has further improved the clarity of the results and the discussion section.
Comment 7: The conclusion should be more concise, to summarize and bring into context with the impact on society. The recommendations and other discussions should become part of the main body of text.
Response 7: Thank you for pointing this out. In response, we have revised the conclusion to be more concise and focused. It now provides a brief summary of the key findings, places the results in a broader societal context, and emphasises the study’s potential impact. Additionally, the revised conclusion is approximately half the length of the previous version, ensuring a more focused and effective closing to the manuscript. Please see the revised conclusion on page 14, lines 505-527.
Reviewer 2 Report
Comments and Suggestions for Authors
Dear Authors,
I kindly ask you to make your results more perceptive. Namely, table 4 and 5 must be graphically interpreted using the histograms per each Food safety characteristics with explanation of observation number in legend of each chart or in the chart title below. Thank you.
Also, I kindly ask you to replace the tables 6 and 7 of PCA with the charts of PCA. Tables are completely unperceptive for interpretation of PCA analysis. Thank you.

Author Response
Please see our response to Reviewer 2's comments attached. As we have included figures in our response, we opted to send it in the attached document. Thank you.

Reviewer 3 Report
Comments and Suggestions for Authors
A very interesting and well-done study. It addresses an important topic that is rarely researched.
Abstract:
Make it more interesting. Suggestion: The studied ‘‘Master Chef cooking show’’ has been featured in many TV broadcasts in different countries with different social and cultural backgrounds. To illustrate this the article presents and compares two different countries – UK and Turkey – with different cultural and culinary traditions.
In general:
However, the article is quite long. It would benefit from being divided into two articles; one “food safety violations’’ and one on ‘‘behaviours for preventing food waste’’. The results section is long and contains more than results. The discussion, however, is thin and have unnecessary repetitions of some of the results. The article would benefit from merging Results and Discussion.
Author Response
Comment 1: A very interesting and well-done study. It addresses an important topic that is rarely researched.
Response 1: Thank you very much for this encouraging comment.
Comment 2: Abstract: Make it more interesting. Suggestion: The studied ‘‘Master Chef cooking show’’ has been featured in many TV broadcasts in different countries with different social and cultural backgrounds. To illustrate this the article presents and compares two different countries – UK and Turkey – with different cultural and culinary traditions.
Response 2: Thank you for your comment and suggestion, which helped us improve the clarity and impact of the abstract while making it more engaging.
In response to your comment and a similar comment from Reviewer 4, we have revised the abstract to better highlight the global relevance of the MasterChef franchise and the social and cultural contrast between the UK and Turkey. The revised version now opens with a broader contextualisation of MasterChef as a globally franchised series that reflects diverse culinary traditions and influences viewers’ practices. We also clarified the comparative nature of the study by explicitly stating the socio-cultural distinctiveness of the two countries examined. Please see the revised abstract marked in red, on p.1, lines 9-25
Comment 3: In general:However, the article is quite long. It would benefit from being divided into two articles; one “food safety violations’’ and one on ‘‘behaviours for preventing food waste’’. The results section is long and contains more than results. The discussion, however, is thin and have unnecessary repetitions of some of the results. The article would benefit from merging Results and Discussion.
Response 3: Thank you for your comments. We acknowledge your concern regarding the overall length and structure of the manuscript.
In response to your suggestion and a similar suggestion from Reviewer 1, we have merged the Results and Discussion sections to make the narrative clearer and more focused, thereby reducing redundancy. This restructuring has allowed us to present the findings more cohesively while integrating interpretation and context throughout.
While we appreciate your suggestion to divide the manuscript into two separate articles (one on food safety violations and one on behaviours for preventing food waste), we believe that the combined analysis is a key strength and novelty of our study.
To further improve clarity and conciseness, we have (1) revised the PCA tables to make them more informative and reader-friendly, (2) moved Tables 4 and 5 (from the previous version) to the Supplementary Information, in line with comments from Reviewers 2 and 4, and (3) condensed the Conclusion section to avoid repetition and better highlight the impact of our study.
We believe that these revisions have enhanced the clarity, focus, and readability of our manuscript, while preserving the integrated approach that underpins our study’s contribution.
Reviewer 4 Report
Comments and Suggestions for Authors
Thank you very much for giving me the opportunity to review the manuscript entitled “Food Safety and Waste Management in TV Cooking Shows: A Comparative Study of Turkey and the UK.” This paper examines how two popular MasterChef series—MasterChef UK and MasterChef Turkey—portray and communicate food safety and waste behaviours. Employing a mixed-methods design, the authors combine video content analysis against predefined safety and waste criteria with transcript-based content analysis and principal component analysis to categorize observed practices. Their findings highlight frequent personal hygiene breaches in the UK version and cross-contamination issues in the Turkish edition, while both programmes feature widespread use of disposables and the discarding of unused food parts. The study further explores how cultural and culinary contexts shape these behaviours. Finally, the authors propose that cooking shows engage food safety and waste management experts during production and more clearly communicate their policies on official platforms, given the influential role these programmes play in shaping viewers’ practices. I appreciate the rigor of the methodological approach and look forward to providing detailed feedback.
Detailed comments of minor significance
Abstract: The abstract could be stronger if it highlighted key quantitative findings (e.g., the percentage of episodes showing personal‐hygiene breaches or cross‐contamination, and the prevalence of disposable use and food‐part waste). A sentence summarizing how Turkish and UK culinary cultures differently shaped these behaviors would sharpen the cross‐national insight. Conclude with concrete implications—such as the recommendation to involve food‐safety experts in production and to communicate waste‐management policies more clearly on official platforms—to make the take‐home message unmistakable. Finally, briefly noting the study’s novelty as the first cross‐national, mixed‐methods analysis of food‐safety and waste behaviors in prime‐time cooking shows will underscore its unique contribution to both media studies and sustainability education.
Materials and methods: While Video Content Analysis (VCA) and traditional Content Analysis (CA) can be powerful tools for systematically cataloguing observed behaviors, they also introduce several challenges: their reliance on predefined categories and observable events can oversimplify complex, context-dependent practices. Robust intercoder training, triangulation with viewer surveys or interviews, and careful attention to video-editing conventions are essential to mitigate these limitations. How were these mitigation strategies introduced and employed in the present study?
The current tables are dense and not reader-friendly. I recommend restructuring them.
Author Response
Thank you very much for giving me the opportunity to review the manuscript entitled “Food Safety and Waste Management in TV Cooking Shows: A Comparative Study of Turkey and the UK.” This paper examines how two popular MasterChef series—MasterChef UK and MasterChef Turkey—portray and communicate food safety and waste behaviours. Employing a mixed-methods design, the authors combine video content analysis against predefined safety and waste criteria with transcript-based content analysis and principal component analysis to categorize observed practices. Their findings highlight frequent personal hygiene breaches in the UK version and cross-contamination issues in the Turkish edition, while both programmes feature widespread use of disposables and the discarding of unused food parts. The study further explores how cultural and culinary contexts shape these behaviours. Finally, the authors propose that cooking shows engage food safety and waste management experts during production and more clearly communicate their policies on official platforms, given the influential role these programmes play in shaping viewers’ practices. I appreciate the rigor of the methodological approach and look forward to providing detailed feedback.
Comment 1: Detailed comments of minor significance. Abstract: The abstract could be stronger if it highlighted key quantitative findings (e.g., the percentage of episodes showing personal‐hygiene breaches or cross‐contamination, and the prevalence of disposable use and food‐part waste).
Response 1: In response to your suggestions and Reviewer 3’s comments, we have revised the abstract to make it more engaging and informative.
Regarding the inclusion of key quantitative findings, we have clarified these in the revised manuscript, as below (also see pp. 8, lines 287–289, and 10, lines 377–383). However, we presented the number of observations per time interval instead of percentages because the durations of MasterChef UK and MasterChef Turkey episodes differed significantly (i.e., episodes from the Turkish version are substantially longer). Therefore, we opted not to provide these figures in the abstract to avoid potential confusion and maintain clarity for readers. However, if you advise us to include the number of observations per time interval (as below) in the abstract, we would be happy to do this.
|
“Personal hygiene violations were observed more frequently on MasterChef UK, occurring every 57 seconds, compared to every 2 minutes and 32 seconds on MasterChef Turkey” [p.8, 287-289] “MasterChef UK recorded 91 relevant observations every 4 minutes and 40 seconds, while MasterChef Turkey recorded 158 observations every 7 minutes and 35 seconds in response to the question” [p.9, lines 333-335] |
Comment 2: A sentence summarizing how Turkish and UK culinary cultures differently shaped these behaviors would sharpen the cross‐national insight.
Response 2: Thanks for this comment. To address your comment and Reviewer 1’s suggestion regarding a concise introduction and sharpening the cross-national differences, we have revised the introduction. We have addressed your comment specifically on p.3, lines 63-79 in the revised manuscript.
Comment 3: Conclude with concrete implications—such as the recommendation to involve food‐safety experts in production and to communicate waste‐management policies more clearly on official platforms—to make the take‐home message unmistakable.
Response 3:Thanks for pointing this out. To address the similar comments and suggestions from you and Reviewer 1 about the conclusion section, we have rewritten the conclusion. Our revision includes more explicit implications, as well. Please see page 14, lines 506-524.
Comment 4: Finally, briefly noting the study’s novelty as the first cross‐national, mixed‐methods analysis of food‐safety and waste behaviors in prime‐time cooking shows will underscore its unique contribution to both media studies and sustainability education.
Response 4: Thank you for your suggestion. In response, we have incorporated a statement highlighting the study’s novelty as the first cross-national, mixed-methods analysis of food safety and waste behaviours in prime-time cooking shows. This addition has been included in the revised conclusion, as below (also see p.14, lines 514–517).
|
As the first cross-national and mixed-methods analysis of food safety and waste behaviours in a prime-time cooking show, our research makes a novel contribution to both media studies and sustainability education. It offers a new perspective for understanding the cultural and social dynamics of televised food media |
Comment 5: Materials and methods: While Video Content Analysis (VCA) and traditional Content Analysis (CA) can be powerful tools for systematically cataloguing observed behaviors, they also introduce several challenges: their reliance on predefined categories and observable events can oversimplify complex, context-dependent practices. Robust intercoder training, triangulation with viewer surveys or interviews, and careful attention to video-editing conventions are essential to mitigate these limitations.
How were these mitigation strategies introduced and employed in the present study?
Response 5: Thanks for your comment regarding the methodological challenges associated with Video Content Analysis (VCA) and Content Analysis (CA). We acknowledge the potential limitations of relying on predefined categories and observable behaviours, and we took several steps to enhance the rigour and reliability of our analysis.
To address concerns related to coding reliability, two independent coders analysed the same video segments and compared their observations. This process helped ensure consistency and reduce subjectivity. Details of this procedure are included in the revised manuscript, as below (also see page 7, lines 255–261 in the revised manuscript).
For the transcript analysis, the first author, who has over ten years of experience in qualitative research, led the process using MAXQDA PRO, a professional software tool designed to support systematic coding and analysis that we purchased specifically to support this study In addition, all analytical procedures were conducted under the supervision of the third author, who has over 20 years of experience in qualitative research.
While we did not incorporate triangulation with viewer surveys or interviews due to the scope of this study, we recognise its value and consider it a promising direction for future research.
|
An inter-rater reliability analysis was conducted with NVivo 14 to evaluate the consistency of observations within the video content analysis. Two independent coders assessed the same video segments, and their observations were compared. For food safety behaviours, the analysis yielded an average Cohen’s kappa value of 0.94 and an agreement rate of 99.95%. For waste behaviours, the average kappa value was 0.91, with an agreement rate of 99.91%. A kappa value of 0.70 or higher is generally considered acceptable as an indicator of inter-rater reliability [56]. |
Comment 6: The current tables are dense and not reader-friendly. I recommend restructuring them.
Response 6: Thanks for pointing this out. In response to your comment and suggestions of Reviewer 2, we have relocated Tables 4 and 5 (which were in the original manuscript) to the Supplementary Information and revised their formatting for improved clarity. We have also revised Tables 3 and 4, which are now located on pages 7 and 11-12 of the revised manuscript. These tables include additional information, such as the percentage of variance explained and the cumulative percentage of variance for each principal component. We have also included the insignificant factor loadings for transparency.
Round 2
Reviewer 2 Report
Comments and Suggestions for Authors
Dear Authors,
The improvements in second version of your manuscript are quite visible and the explanations you gave in your cover letter/response to reviewers are quite acceptable. Thus, I will suggest your paper to be published. Thank you.
Author Response
Comment 1:
Dear Authors,
The improvements in second version of your manuscript are quite visible and the explanations you gave in your cover letter/response to reviewers are quite acceptable. Thus, I will suggest your paper to be published. Thank you.
Response 1:
We thank the Reviewer for their helpful comments and suggestions regarding our original submission. They helped us a lot to improve our manuscript for the second round of reviews. We are pleased to hear that the Reviewer accepted our responses and revisions and recommended our paper for publication.